# Particle Flow Analysis on Mechanical Characteristics of Rock with Two Pre-Existing Fissures

Zhenzi Yu [1,2], Ang Li [3,*], Bo Zhang [1,2], Hongyue Li [4], Qian Mu [5], Yonggen Zhou [3] and Shuai Gao [3]

1   State Key Laboratory of Coking Coal Exploitation and Comprehensive Utilization, China Pingmei Shenma Group, Pingdingshan 467099, China
2   Institute of Coal Mining and Utilization Pingdingshan Tianan Coal Mining Co., Ltd., Pingdingshan 467099, China
3   School of Architecture and Civil Engineering, Xi'an University of Science and Technology, Xi'an 710054, China
4   No.12 Mine, Pingdingshan Tian'an Coal Mining Co., Ltd., Pingdingshan 467000, China
5   CCTEG Chongqing Research Institute, Jiulongpo, Chongqing 400039, China
*   Correspondence: angli@xust.edu.cn

**Abstract:** Many research results show that under any stress state the rock mass is most likely to crack, swell, bifurcate, and infiltrate from the fissure tip, resulting in rock engineering instability and failure. In order to study the influence of double fissure angles on rock mechanical characteristics, five rock numerical models with different fissure angles were established by numerical simulation software. Uniaxial compression tests were carried out, and the variation characteristics of rock stress, strain, failure, microcrack, and acoustic emission (AE) were recorded. The test results show that: With increases in the fissure angles, the elastic modulus of rock increased, while the peak strength decreased first and then increased. The number of microcracks in rock was greater at 15° and 75° than at other angles. The microcracks in rock were mainly tensile cracks, and relatively few were shear cracks. The angles of microcracks were mostly concentrated between 0 and 180°, most of which were between 60 and 110°. The failure of rock was relatively light when the fissure angle was 15° or 75°, but it produced more and smaller fragments, and the failure was the most serious when the fissure angle was 30°. The angles of the fissures affected the maximum number of AE events, the strain values for the initial AE event, and the maximal AE event. This research can provide some reference for disasters caused by rocks with pre-existing fissures.

**Keywords:** pre-existing fissures; uniaxial compression; microcrack; failure characteristics; acoustic emission

## 1. Introduction

Rock has become a geological body that must be dealt with in rock engineering, such as mining engineering, tunnel engineering, carbon dioxide geological storage, nuclear waste storage, and geothermal mining [1–5]. As a naturally formed geological body, rock mass is composed of various joint fissures and rock blocks cut by fissures. Many research results show that under any stress state rock mass is most likely to crack, expand, bifurcate, and penetrate from the fissure tips, resulting in instability and failure [6–8]. Therefore, it is of great significance to study the basic mechanical characteristics, failure mechanism, and mechanical model of fissures for the safety and stability of rock engineering.

Fissures include primary fissures and secondary fissures. Primary fissures are formed during diagenesis, and secondary fissures are formed by external forces after rock diagenesis [9]. Domestic scholars have conducted a lot of research on the influence of fissures on rocks, including large-scale in situ tests and laboratory-scale fissure research [6–8]. Among them, most of the fissure research in the laboratory involves taking rock samples with a height/diameter ratio of 2:1, according to the standards of the international society of rock mechanics, and then forming fissures through hydraulic cutting to study their influence

on rock strength, deformation, and failure. Wong and Chau [10] studied the influence of fissure angles, bridge angles, and friction coefficients on the crack combination modes and strength of rock specimens with two parallel fissures and determined three typical crack combination modes. Wong and Einstein [11] studied the influence of fissure inclination and rock bridge angles on crack growth under uniaxial loading and found nine types of crack coalescence. Cao et al. [12] prepared rock samples with two fissures and carried out uniaxial compression tests to determine seven types of crack coalescence. Lee and Jean [13] conducted uniaxial compression tests on granite containing two nonparallel fissures (one horizontal and one inclined). The results showed that tensile cracks of granite are always accompanied by the germination of shear cracks. Zhou et al. [14] investigated the types of crack coalescence in specimens with multiple fissures and found another two crack types. These studies on the influence of one/two fissures on the mechanical properties and crack evolution of rock/rock-like material provide a good basis for studying the mechanical properties of rock containing two parallel fissures.

Laboratory test research can usually only obtain the shape of crack development and cannot explain its micromechanism well, so the theoretical and numerical analysis research of fissured rock has gradually developed and expanded. In addition, during numerical simulation, only one geometric feature of the fissure can be changed, while other factors remain unchanged, to study its influence on rock mechanical properties. In laboratory tests, even in the same rock or rock-like standard sample, its homogeneity will be different [15,16]. At present, the numerical methods for studying fissured rock can be summarized into three categories: the continuous medium analysis method, the discontinuous analysis method, and the mixed analysis method. Among them, the discrete element method proposed by Cundall [17] has unique advantages in simulating discontinuous medium materials such as rocks. The nonlinear deformation and crack development characteristics in a jointed rock mass can be simulated more realistically by the discrete element method, and the discrete element software PFC2D has the advantage of counting the direction and number of microcracks in the rock and can better master crack initiation and propagation. Zhang and Wong [18] used the two-dimensional discrete element program PFC2D to simulate the law of crack initiation, propagation, and through failure of a gypsum sample with pre-set fissures under uniaxial load and successfully used the same program to simulate the process of bridging and through failure of multiple pre-set fissures in the gypsum sample. Lin et al. [19] simulated a uniaxial compression test of a double-hole jointed rock mass through discrete element modeling and obtained its strength and failure characteristics. Jin et al. [20] analyzed the influence of pre-set fissures on the crack initiation and failure mode of a specimen under uniaxial compression from the perspective of energy. Based on the particle flow theory, Zhuang [21] studied the law of crack propagation and initiation in a fractured rock mass from the perspective of micromechanics using PFC2D and derived the formula of crack propagation and initiation using macrofracture mechanics and other theories. The above results show that PFC numerical simulation can be used to analyze the fracture behavior of rock or rock-like materials with fissures.

A lot of work has been carried out on the influence of single/double fissures on the mechanical properties and failure modes of rocks. However, the relationship between microcrack evolution and AE event behavior under the influence of various factors (including fissure angle) is still limited and needs further research. Therefore, on the basis of previous studies, this paper considers the simultaneous change in angle of double through fissures and studies its influence on rock strength, deformation, failure, and AE characteristics. At the same time, using the advantages of PFC2D numerical simulation software, we focus on the influence of double through fissures on the change in microcracks and try to explain the influence of fissure angle on rock mechanical characteristics from a micro perspective. The research results can better guide the construction and disaster control of rock engineering with pre-existing fissures.

## 2. Numerical Model Construction

### 2.1. Parallel Bonding Model

As a discrete element method software, PFC is suitable for studying the fracture and fracture development of particle aggregates. PFC attempts to explain the mechanical properties and behavior of media from a meso perspective, which has been widely used in rock engineering [17,17,22]. The bonding between particles is damaged by external effects, resulting in the separation of particles to simulate the generation and propagation of cracks in the medium. In the process of simulating particle bonding failure, the PFC program provides two basic particle bonding models: contact bonding and parallel bonding [17,17,22]. The particle models of the two kinds of bonding and their micromechanical behavior are shown in Figure 1. However, the contact bonding model uses point contact and cannot transmit torque, so the parallel bonding model is used more often in rock simulation. The normal stress and tangential stress on parallel bonding are expressed by the following formulas:

$$\bar{\sigma} = \frac{-\bar{F}_i^n}{A_2} + \frac{\left|\bar{M}_i^s\right|\bar{R}}{I}$$

$$\bar{\tau} = \frac{-\bar{F}_i^s}{A_2} + \frac{\left|\bar{M}_i^n\right|\bar{R}}{J}$$

(1)

where $A_2$ is the area of the parallel bonding section, $J$ is the polar moment of inertia of the section, and $I$ is the moment of inertia of the section in the direction of rotation along the contact point. $\bar{R}$ is the bonding radius, and $\bar{F}_i^n$ and $\bar{F}_i^s$ are the normal and tangential components of the force after bonding. $\bar{M}_i^n$ and $\bar{M}_i^s$ are the normal and tangential components of the bending moment after bonding. When the normal or tangential stress exceeds the corresponding parallel bonding strength, the parallel bonding failure will produce tensile microcracks or shear microcracks, respectively.

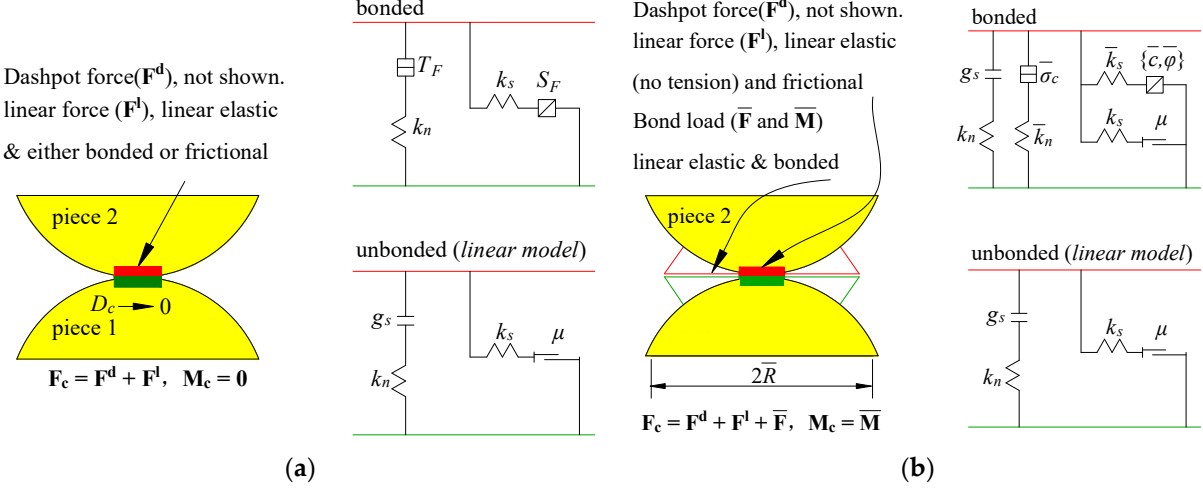

(a)　　　　　　　　　　　　　　(b)

**Figure 1.** Bonded-particle model and its micromechanical behavior in PFC numerical simulation [17,17,22]. (**a**) Contact bond model. (**b**) Parallel bond model.

### 2.2. Parameters of Numerical Rock Models

In PFC simulation, the macroscopic mechanical properties of a rock model are determined by the microscopic mechanical properties of particles and bonds. However, these microscopic parameters cannot be directly derived from field and laboratory tests. Before numerical simulations are performed, the selection and verification of microscopic parameters is required. Typically, the microscopic parameters of a PFC rock model are

calibrated by simulating uniaxial compression experiments. During the calibration process, the microscopic parameters of the particles and bonds are adjusted many times by "trial and error" until these parameters can better reflect the mechanical properties of the real rock [17,17,22]. The load of the numerical model is applied by moving the upper wall at a loading rate of 0.05 mm/s. All conditions of the numerical test are the same as the laboratory test conditions. Through the "trial and error method" of repeated inspection and comparison, the physical and mechanical parameters listed in Table 1 can accurately reflect the macroscopic mechanical properties of real sandstone. The stress–strain curves and failure modes of the PFC model (Figure 2a) are in good agreement with the laboratory results of real sandstones. Based on the uniaxial compression laboratory tests of sandstone, three PFC numerical models were established, as shown in Figure 2b.

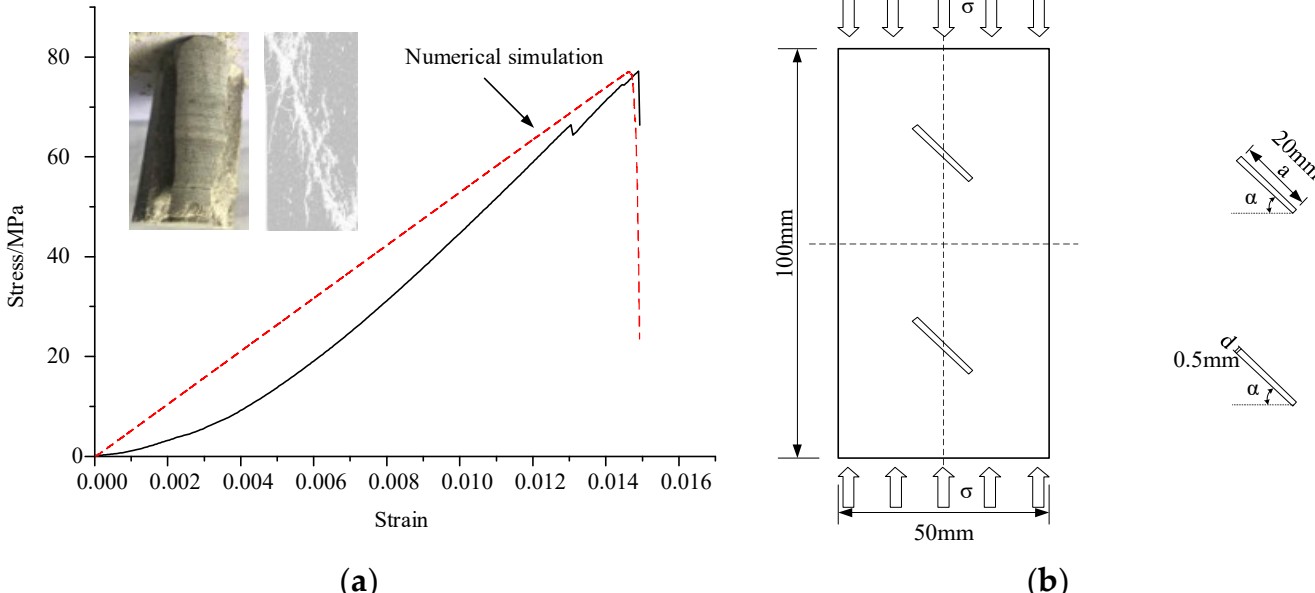

**Figure 2.** Stress–strain curves, failure modes, and numerical model in uniaxial compression experimental tests and numerical simulations. (**a**) Stress–strain curves and failure modes. (**b**) Numerical model based on PFC.

**Table 1.** Micromechanical parameters of rock in the numerical simulation.

| Parameters | Value | Parameters | Value |
|---|---|---|---|
| Minimum particle size ($R_{min}$) | 0.2 | Ratio of normal to tangential bonding contact stiffness ($\bar{K}_n / \bar{K}_s$) | 1.5 |
| Ratio of maximum particle size to minimum particle size ($R_{max} / R_{min}$) | 1.5 | Average and standard deviation of normal bond strength ($\sigma_b$) | 16 |
| Effective modulus of particles ($E_c$) | 1.8 | Mean and standard deviation of cohesive force ($c_b$) | 20 |
| Ratio of the contact stiffness between the normal direction and the tangential bond of particles ($K_n / K_s$) | 1.5 | Bond internal friction angle ($\phi$) | 42 |
| Bond effective modulus ($\bar{E}$) | 2.4 | Linear friction coefficient of particles ($\bar{\mu}$) | 0.5 |

### 2.3. Numerical Rock Models with Two Pre-Existing Fissures of Different Angles

In order to study the mechanical characteristics of rock with two pre-existing fissures of different angles, five numerical rock models with the same L (length) and d (width) values were built with different α (angle) values, as shown in Figure 3.

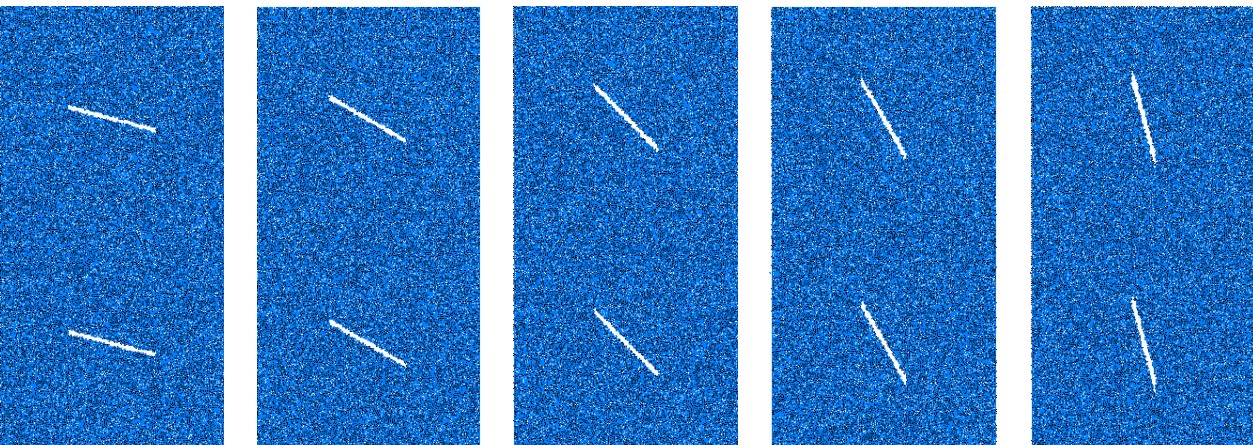

**Figure 3.** Numerical models and loading conditions of rock samples under uniaxial compression.

### 2.4. Acoustic Emission Simulation by PFC

Under the influence of external load, when the stress intensity transmitted between particles exceeds the bond strength between particles, bond fracture produces microcracks in rock samples [23]. When the microcracks propagate in a rock sample, the damage energy is rapidly released in the form of sound waves, that is, the phenomenon of acoustic emission (AE) [24,25]. Therefore, AE events can be simulated by calculating the number of particle bond breaks during numerical experiments. Due to the limitation of computing power, the particle size and particle number of PFC2D cannot directly reach the mechanical response level of real macroscopic rocks, but the reflected mechanical laws are helpful for understanding the AE phenomenon of rocks [26–28].

## 3. Numerical Simulation Results

### 3.1. Strength and Deformation Characteristics

Figure 4 shows the stress–strain curves, elastic modulus, peak strain, and peak strength of different fissure dip angles. It can be seen from Figure 4a that the variation trends of stress–strain curves of rocks with different dip angles were basically the same. The difference was that with the change in dip angles the changes in the elastic modulus, peak strain, and peak strength were different. With increases in fissure dip angles, the elastic modulus increased from 3.74 GPa at 15° to 4.91 GPa at 75°, an increase of 31.3% (Figure 4b). The peak strength first decreased and then increased with increases in angle, from 45.25 MPa at a 15° angle to 41.66 MPa at a 30° angle, and then increased to 53.57 MPa at 75° (Figure 4c). The peak strain had no fixed law with an increase in angle, but it increased gradually overall (Figure 4d).

### 3.2. Microcrack Evolution Characteristics

The change trend of the number of cracks in rock at different fissure angles is shown in Figure 5a. The change in crack number and the change in the stress–strain curves are shown in Figure 5b (with limited space, only the case with an angle of 45° is shown). The change in microcracks with strain can be roughly divided into three stages, as shown in Figure 5b. When the stress is small, there are basically no microcracks. Then, microcracks gradually occur with the increase in stress, and a large number of rock fracture microcracks occur at the peak strength of the rock. The number of microcracks at the peak strength is not the most in the whole process, but it is a turning point with a large number of microcracks.

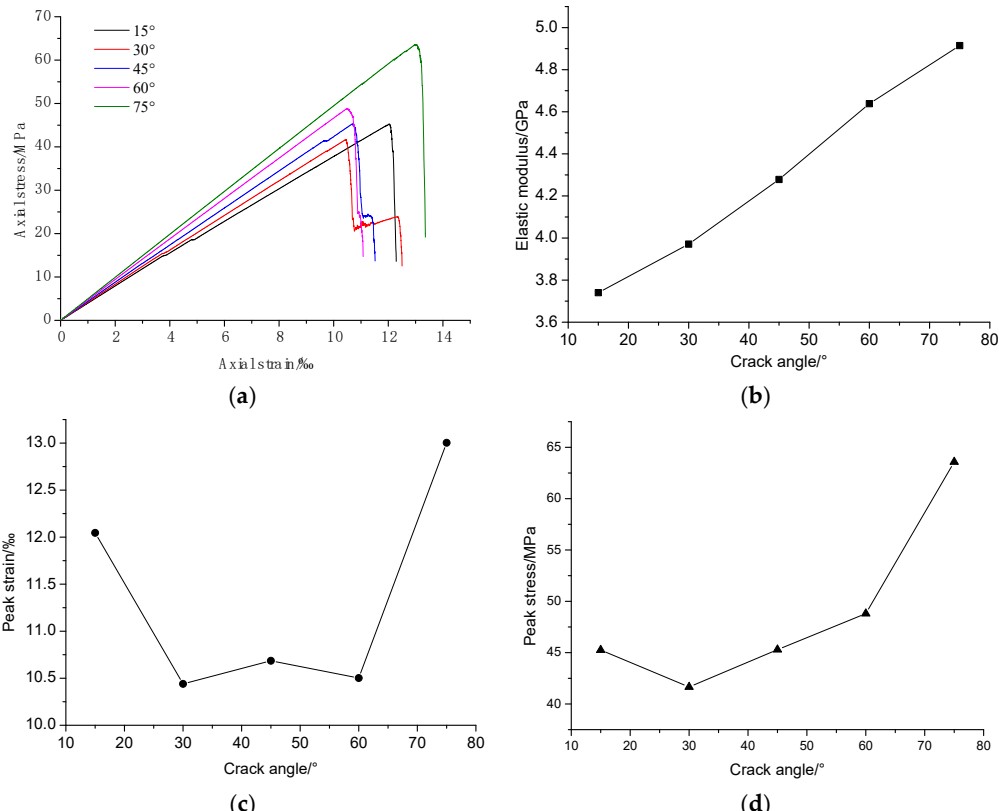

**Figure 4.** Strength and deformation characteristics with different fissure angles under uniaxial compression. (**a**) Stress–strain curves. (**b**) Elastic modulus. (**c**) Peak strain. (**d**) Peak strength.

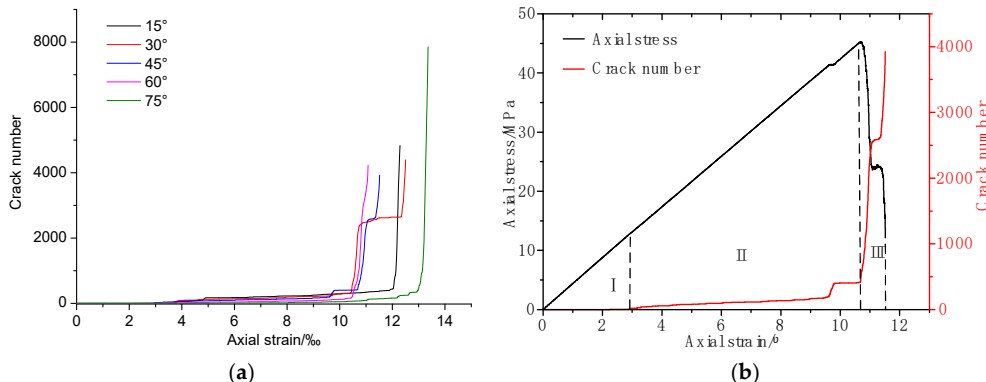

**Figure 5.** Microcrack evolution with axial strain at different fissure angles under uniaxial compression. (**a**) Microcrack evolution with axial strain. (**b**) Microcrack evolution at 45°.

In the process of uniaxial compression, the total number of microcracks and the number of cracks at the peak strength are shown in Figure 6. The total number of microcracks decreased first and then increased with increases in the fissure angles. The minimum number was 3926 when the angle was 45°. Compared with 4832 cracks at the angle of 15°, the number of cracks decreased by 18.8%, and when the angle increased to 75°, the number of cracks was 7851, which was an increase of 100% compared with 45°. However, the number of microcracks at the peak strength had no fixed law with the change in angle.

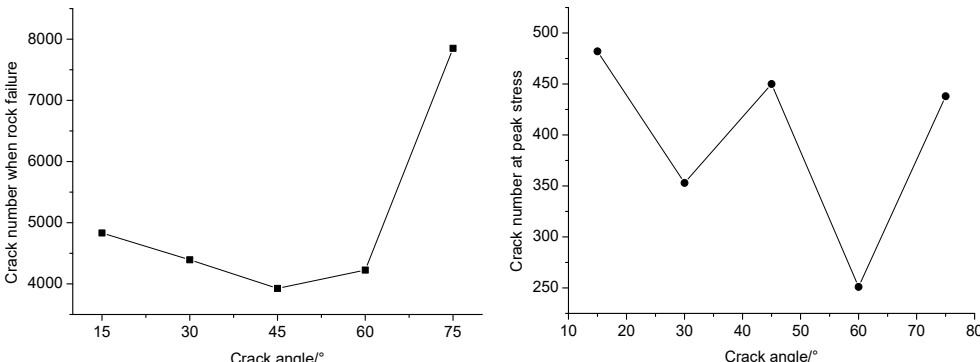

**Figure 6.** The total number of microcracks and the number of cracks at the peak strength under uniaxial compression.

In order to further study the evolution characteristics of microcracks. Figure 7 shows rose diagrams of rock microcracks at different angles of fissures. The angles of microcrack generation were concentrated between 0 and180°, most of which were between 60 and 110°. With increases in the angle, the 90° microcracks decreased first and then increased, from 830 at the fissure angle of 30° to 693 at the angle of 45° and then to 1419 at the angle of 75°. That is, when the fissure angle was 15° or 75°, more 90° microcracks were generated.

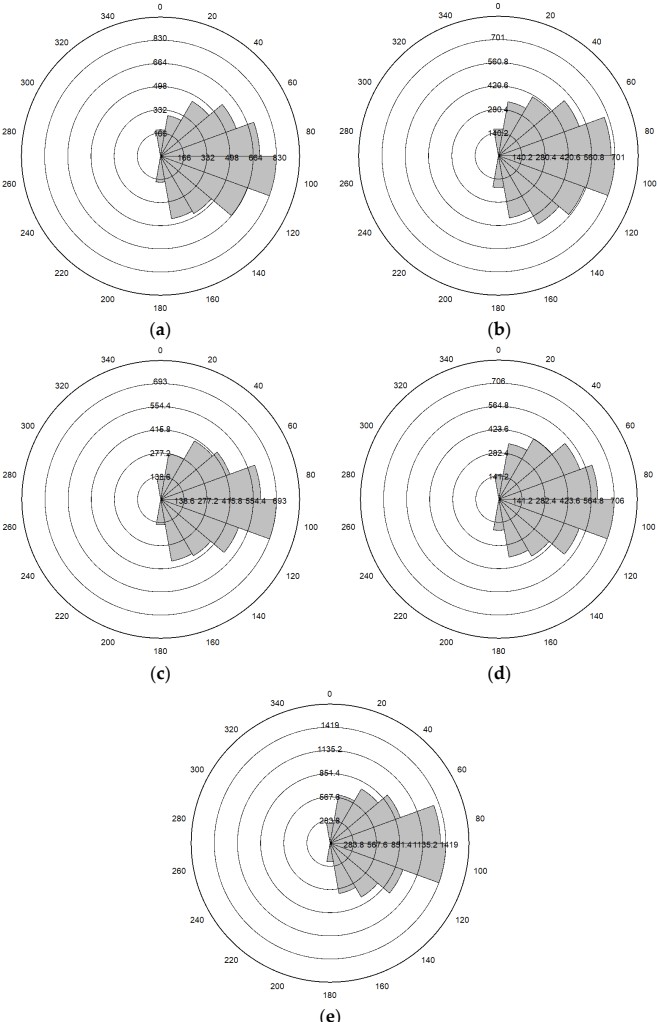

**Figure 7.** Rose diagrams of rock microcracks with different angles of pre-existing fissures under uniaxial compression: (**a**) 15°, (**b**) 30°, (**c**) 45°, (**d**) 60°, (**e**) 75°.

Figures 8 and 9, respectively, show the distribution of tensile cracks and shear cracks during rock failure at different dip angles. It can be seen that the microcracks of rock during uniaxial compression were mainly tensile cracks. Tensile cracks did not occur only at the fissure tip. At the distance of the fissure tip, tensile crack lines also occurred. Shear cracks mostly occurred at the fissure tip. In addition, both tensile and shear cracks increased with increases in angle.

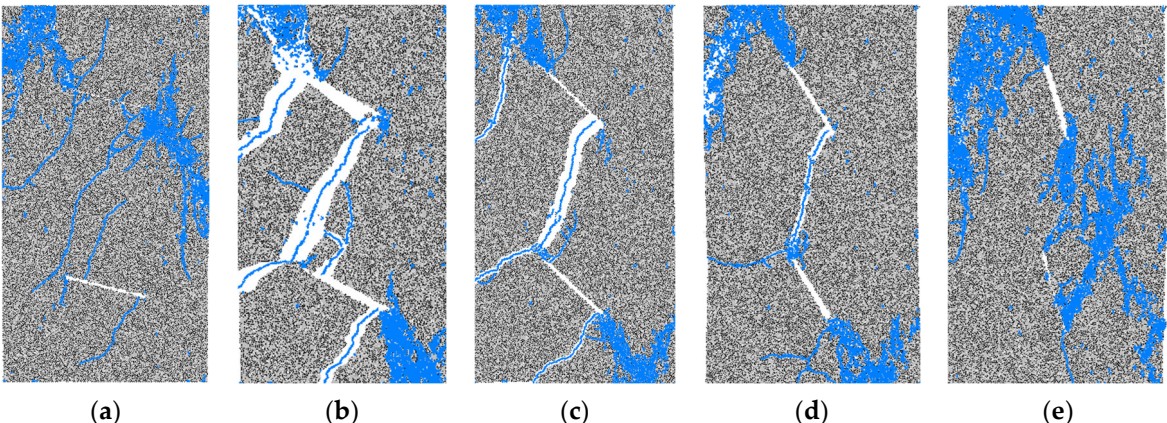

(**a**)      (**b**)      (**c**)      (**d**)      (**e**)

**Figure 8.** The distribution of tensile cracks during rock failure with different fissure angles under uniaxial compression: (**a**) 15°, (**b**) 30°, (**c**) 45°, (**d**) 60°, (**e**) 75°.

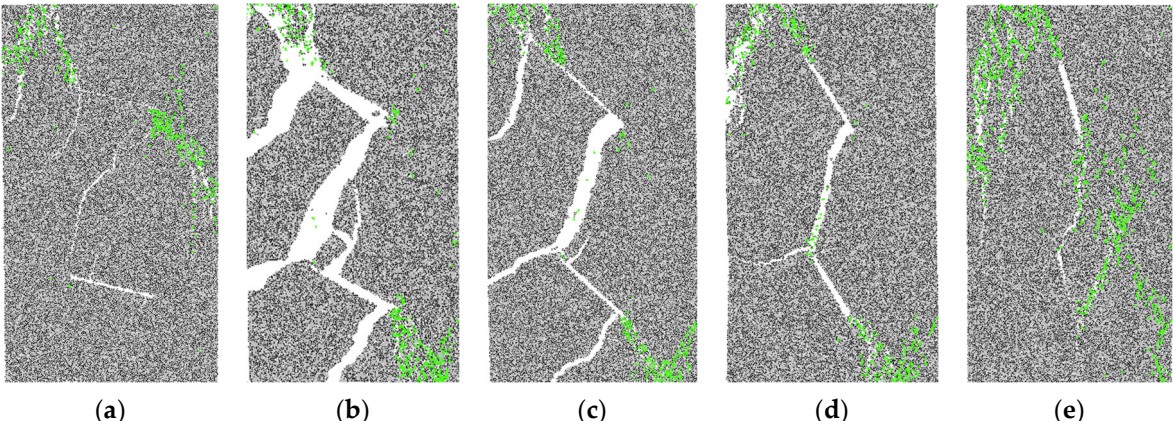

(**a**)      (**b**)      (**c**)      (**d**)      (**e**)

**Figure 9.** The distribution of shear cracks during rock failure with different fissure angles under uniaxial compression: (**a**) 15°, (**b**) 30°, (**c**) 45°, (**d**) 60°, (**e**) 75°.

*3.3. Failure Modes*

The fissure dip angle has a great influence on the failure modes of rock. As can be seen from Figure 10, when the angles are 15° and 75° the rock fracture was relatively light, but the failure produced more fragments. It can also be explained from Figure 5 that when the angle was 15° or 75° there were more microcracks in the rock than at other angles as well as more cracks and more energy dissipation. Therefore, during rock failure, although the rock fracture was not violent, the fragments produced were relatively small and greater in number. When the angle was 30°, the rock was broken most seriously. The fracture not only ran between the two fissures but also ran through the whole rock due to the failure of the cracks. The failure modes of 45° and 60° were basically the same as those at 30°, but the degree of fracture was smaller than at 30°.

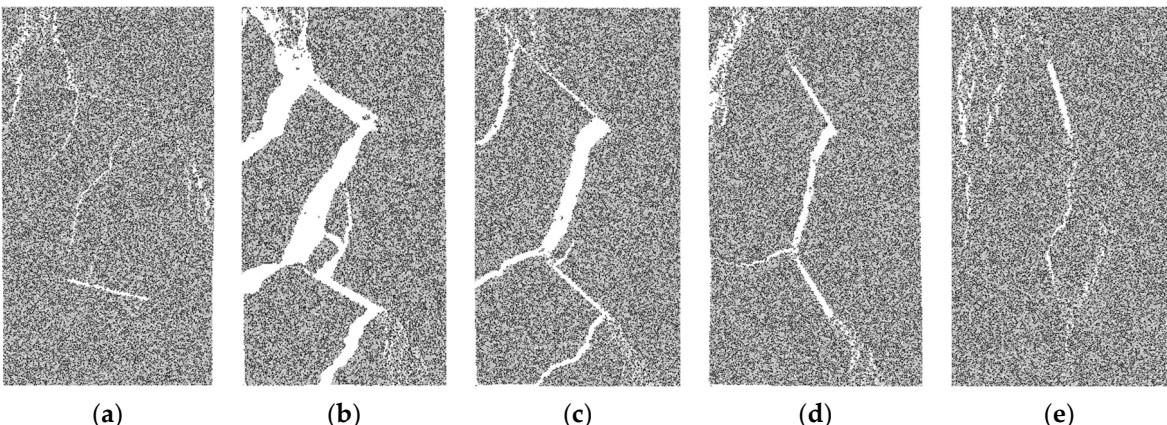

**Figure 10.** The failure modes of rock with different fissure angles under uniaxial compression: (**a**) 15°, (**b**) 30°, (**c**) 45°, (**d**) 60°, (**e**) 75°.

### 3.4. Acoustic Emission Characteristics

Figure 11 shows the curves of the stress–strain AE events of rocks with different α values. It can be seen from the figures that the evolution characteristics of the AE events were closely related to the stress–strain relationship. Before the peak intensity of the stress–strain curve, the number of AE events was very small. When the stress–strain behavior was in the failure stage, the number of AE events peaked and declined rapidly, which means that the rock with two pre-existing fissures was severely damaged in this stage.

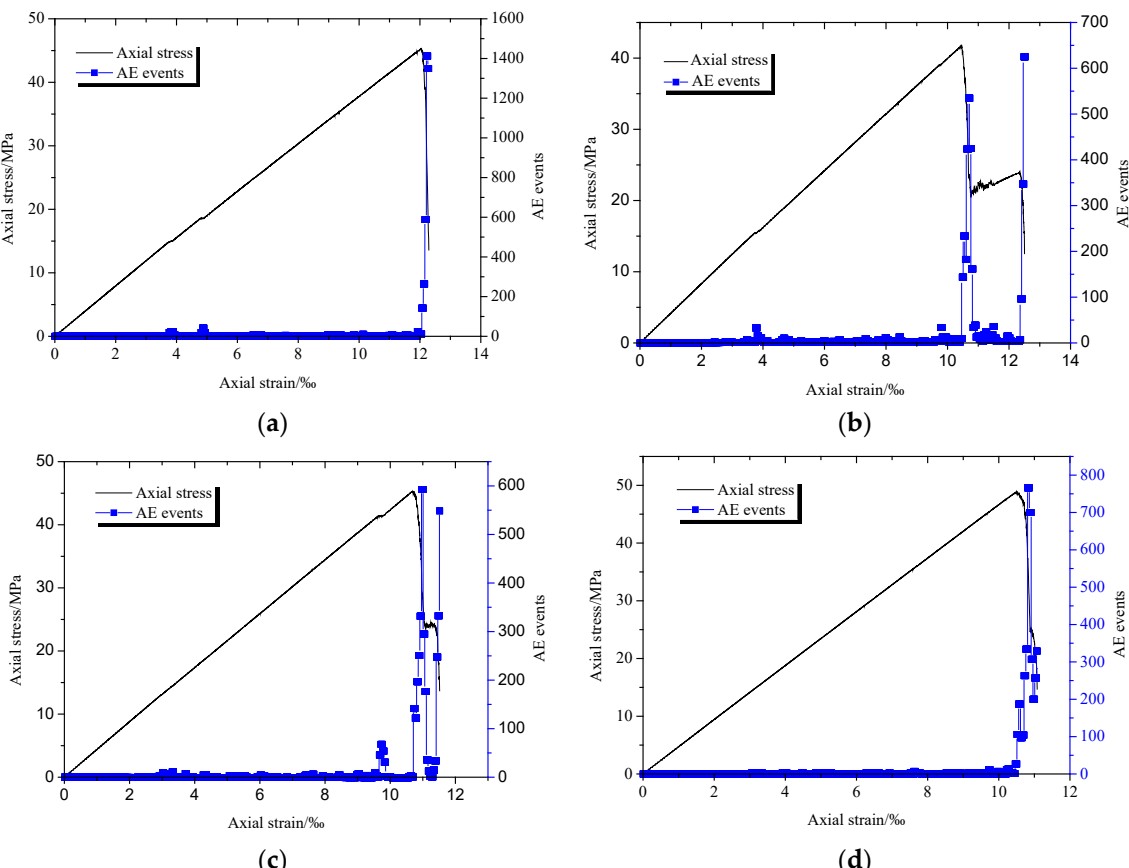

**Figure 11.** *Cont.*

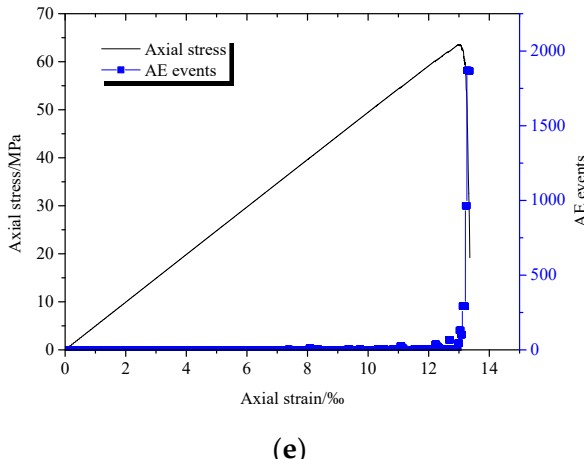

(**e**)

**Figure 11.** Stress–strain AE event curves of numerical rocks with different angles under uniaxial compression: (**a**) 15°, (**b**) 30°, (**c**) 45°, (**d**) 60°, (**e**) 75°.

The effects of different $\alpha$ values on the AE characteristics of rocks are described as follows.

(1) The angle of the fissures affected the maximum number of AE events. As the rock fracture angle increased from 15° to 75°, the maximum AE events decreased first and then increased. When the angle increased from 15° to 45°, the maximum number of AE events decreased from 1453 to 602. The maximum number of AE events decreased by 58.6%. When the angle increased from 45° to 75°, the maximum number of AE events increased from 602 to 1896. The maximum number of AE events increased by 68.2. The evolution characteristics of the maximum number of AE events with different $\alpha$ values were similar to the changes in the peak strength of rocks with two pre-existing fissures.

(2) It affected the strain values for the initial AE event and the maximal AE event. As $\alpha$ increased from 15° to 75°, the strain values of the initial AE event were 3.84‰, 2.46‰, 2.23‰, 2.24‰, and 7.37‰ and the strain values of the maximum AE event were 12.12‰, 12.41‰, 10.93‰, 10.86‰, and 11.51‰.

(3) It affected the strain range for severe AE events near the peak strength. The strain range of severe AE events near the peak strength increased first and then decreased with an increase in $\alpha$. When the angle was 30°, the strain influence range was the largest: 10.51 to 12.42. At the angles of 15° and 75°, the strain influence range was the smallest, and the strain rapidly decreased and disappeared when the peak strength failure was reached.

## 4. Discussion

In this paper, the influence of fissure angles on rock strength, deformation, failure, microcrack evolution, and AE characteristics was studied by using the discrete element simulation software PFC. The elastic modulus increased with an increase in the fissure angle. The peak strength decreased first and then increased. As the rock fracture angle increased from 15° to 75°, the maximum AE events decreased first and then increased. These change characteristics were basically consistent with previous studies [5,10,14]. The difference is that the rose diagrams of the microcrack distributions were obtained in this paper. They can accurately describe the failure directions of microcracks and the number of microcracks in that direction. In the control of engineering the fracture and failure of rock with pre-existing fissures, they can be controlled according to the propagation direction of a large number of microcracks. In this way, the waste of manpower and material resources caused by the control technology, such as grouting or bolting due to a lack of a clear control direction, can be avoided.

However, the research of this paper also has some limitations. In numerical simulation, rock is regarded as an isotropic material. In addition, when other geometric dimensions of

the fissures and loading conditions change, the strength, deformation, microcrack evolution, and other characteristics of the rock change. In addition, the information to employ the rose diagrams of microcrack distribution effectively, such as an accuracy analysis, is not provided. These will be the focus of our next research work.

## 5. Conclusions

(1) With increases in double fissure angles, the elastic modulus of rock increased gradually. The peak strength of rock decreased first and then increased. The peak strain had no fixed law with the increase in angle, but it increased gradually overall.

(2) There were more microcracks when the angle was 15° or 75°. The most microcracks were produced when the fissure angle was 75°, and the least microcracks were produced when the fissure angle was 45°. No matter the fissure angles, the microcracks of rock under uniaxial compression were mainly tensile cracks, with relatively few shear cracks. The angles of the microcracks were concentrated between 0 and 180°, the majority of which were between 60 and 110°.

(3) When the angle was 15° or 75°, the fracture degree of rock was relatively light, but the fragments produced by failure were greater in number and smaller. When the angle was 30°, the fracture degree was the most serious, with fewer and larger fragments.

(4) The angles of the fissures affected the maximum number of AE events, the strain values for the initial AE event, and the maximal AE event. As the rock fracture angle increased from 15° to 75°, the maximum AE events decreased first and then increased. As $\alpha$ increased from 15° to 75°, the strain values of the initial AE event were 3.84‰, 2.46‰, 2.23‰, 2.24‰, and 7.37‰ and the strain values of the maximum AE event were 12.12‰, 12.41‰, 10.93‰, 10.86‰, and 11.51‰.

This study did not consider the heterogeneity of the rock and other geometric characteristics of fissures. However, these conclusions have certain reference significance for the disaster control of rock engineering with fissures. The information to employ the rose diagrams of microcrack distribution effectively, such as an accuracy analysis, is not provided. In the next step, we will conduct more in-depth research on these limitations.

**Author Contributions:** Formal analysis, S.G.; Funding acquisition, B.Z.; Investigation, Q.M.; Methodology, Y.Z.; Software, H.L.; Writing—original draft, Z.Y.; Writing—review & editing, A.L. All authors have read and agreed to the published version of the manuscript.

**Funding:** This research was funded by the National Natural Science Foundation of China (No. 51874229) and the Natural Science Foundation of Shaanxi Province (2020JZ-52).

**Informed Consent Statement:** Not applicable.

**Data Availability Statement:** Not applicable.

**Acknowledgments:** The authors would like to gratefully acknowledge the reviewers that provided helpful comments and insightful suggestions on a draft of the manuscript.

**Conflicts of Interest:** The authors declare no conflict of interest.

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
