# Peer review of "Particle Flow Analysis on Mechanical Characteristics of Rock with Two Pre-Existing Fissures"

_sustainability, doi:10.3390/su142214862_

Round 1
Reviewer 1 Report
I went through the manuscript titled "Particle flow analysis on mechanical characteristics of rock with two pre-existing fissures" written by Zhenzi Yu and others. The manuscript describes about the numerical modeling using the discrete element method (DEM) performed for simulating the rock deformation process with respect to the two pre-existing parallel fissures. The manuscript lacks appropriate reviews of the existing works, thus the authors failed to explain how their results are important for the common knowledges of rock deformation mechanics. For example, in line 37, "Domestic scholars have done a lot of research on the influence of fissures on rocks, including large-scale in-situ test and laboratory scale fissure research." is stated without any appropriate references. It means that the authors show nothing about the research history of the relevant area. Mainly because of such poor review on the existing works, the this manuscript only succeeded in describing the results of simple 5 numerical experiments without any discussions. The authors described some results, but they did not discuss whether these results matches with existing works or theories. Below I note some other point to be modified, though I gave up to pick them all. [specific concerns] Appropriate figure captions are required. The authors provide too simple figure captions such as "Figure 4. Microcrack evolution with axial strain." However, more detailed explanation is required in general. For example, Figure 4b is for the axial angle for 45 degree, which is explained in the text. This kind of explanation is also required in the figure captions, otherwise, readers have difficulty to understand the figures. l60: PFC, which is developed by ITASCA, can be referred following the guideline provided by the company. The authors should appropriately refer the commercial software under their rules. https://www.itascainternational.com/software/faqs/how-should-i-cite-itasca-software-in-my-publication-or-report l81: Figure 1 is not properly shown. l101: maybe the authors had better to show one typical results (such as shown in Figures 7-9) in Figure 2 to explain how to see the results of the present numerical experiments. l163: Figure 7 requires explanation such as "which domain is the rock, fissure and so on" 7 August
Author Response
Sustainability
Responses to comments
Dear Editors and Reviewers,
Ref.: sustainability-1851625
Title: Particle flow analysis on mechanical characteristics of rock with two pre-existing fissures
Thank you very much for giving us this opportunity to revise our manuscript, and we are very grateful to the editor and reviewers for the helpful suggestions. Those comments are all valuable and very helpful for revising and improving our paper, as well as an important guiding significance to our future researches. We have made revisions as commented, as marked in red in the revised submission.
Together with the revised manuscript, a point-by-point response to the editor’s and reviewers’ comments has been attached.
Thanks for your effort and best regards.
Yours sincerely,
Zhenzi Yu, Ang Li*, Bo Zhang, Hongyue Li, Qian Mu, Yonggen Zhou, Shuai Gao
Detail Revisions for ID sustainability-1851625
Title: Particle flow analysis on mechanical characteristics of rock with two pre-existing fissures
Manuscript Number: sustainability-1851625
Comments of the reviewer #1:
I went through the manuscript titled "Particle flow analysis on mechanical characteristics of rock with two pre-existing fissures" written by Zhenzi Yu and others. The manuscript describes about the numerical modeling using the discrete element method (DEM) performed for simulating the rock deformation process with respect to the two pre-existing parallel fissures.
Question 1: The manuscript lacks appropriate reviews of the existing works, thus the authors failed to explain how their results are important for the common knowledges of rock deformation mechanics. For example, in line 37, "Domestic scholars have done a lot of research on the influence of fissures on rocks, including large-scale in-situ test and laboratory scale fissure research." is stated without any appropriate references. It means that the authors show nothing about the research history of the relevant area.
Response: Thanks for your kind advice. We are sorry for the inconvenience. We rewritten the Introduction according to your opinions and added some relevant references.
Question 2: Mainly because of such poor review on the existing works, this manuscript only succeeded in describing the results of simple 5 numerical experiments without any discussions. The authors described some results, but they did not discuss whether these results matches with existing works or theories.
Response: Thanks for this good suggestion. According to your opinion, we have added some descriptions in the results section and added a discussion section.
Question 3: Appropriate figure captions are required. The authors provide too simple figure captions such as "Figure 4. Microcrack evolution with axial strain." However, more detailed explanation is required in general. For example, Figure 4b is for the axial angle for 45 degree, which is explained in the text. This kind of explanation is also required in the figure captions, otherwise, readers have difficulty to understand the figures.
Response: Thanks for your valuable suggestion. We have revised it according to your opinions. And the similar problems in the full text are checked and modified.
Question 4: PFC, which is developed by ITASCA, can be referred following the guideline provided by the company. The authors should appropriately refer the commercial software under their rules. https://www.itascainternational.com/software/faqs/how-should-i-cite-itasca-software-in-my-publication-or-report.
Response: Thanks for your kind suggestion. We have revised it according to your suggestions.
Question 5: Figure 1 is not properly shown.
Response: Thanks for this good suggestion. We have revised it according to your opinions.
Question 6: maybe the authors had better to show one typical results (such as shown in Figures 7-9) in Figure 2 to explain how to see the results of the present numerical experiments.
Response: Thanks for your kind advice. We did not put Figures 7 to 9 together, mainly because we explained the distribution of shear crack, tensile crack and different m of failure respectively. If they are put into everything, it shows that the tensile crack and the shear crack are somewhat repeated, and it will become difficult to distinguish. This will cause misunderstanding to readers.
Question 7: Figure 7 requires explanation such as "which domain is the rock, fissure and so on".
Response: Thanks for your good suggestion. We are sorry for the inconvenience. We have revised it and marked in red in the revised submission.
Special thanks to you for your good comments. A point-by-point response to the comments has been made, and the revised manuscript shows the changes highlighted in red. All of your suggestions are very important, and they have important guiding significance for my thesis writing and research work. Your efforts in the review process of this manuscript are greatly appreciated.
Reviewer 2 Report
Dear Authors,
The authors presented an approach of “Particle flow analysis on mechanical characteristics of rock with two pre-existing fissures.” The subject is interesting and within the scope of the journal: Sustainability. However, I think significant concerns are reasonable and could improve the manuscript’s quality before publication. Overall, I would recommend publication of this manuscript subject to “major revision” −taking into account precisely addressing the following comments in the revised version.
1. The English language does not comply with the standard required for an international journal article.
2. Abstract has flaws. Please make the abstract precise with the following information: background, methods, results, and concluding remarks.
3. I’d suggest adding model/result validation in the abstract section and the methodology part. Besides, the concluding remarks are not well written. I recommend stressing it in this section.
4. I am not sure what is the main research hypothesis. What are the research gaps?
5. What are the main contributions of this study?
6. Justification of selecting a model is required.
7. Discussion is missing. The discussion section needs to be added separately. The contents sometimes jump between topics without a clear direction. I would have wished to see more information on the actual meaning of the findings and how the results add to the broader issue and the specific scientific field.
Author Response
Sustainability
Responses to comments
Dear Editors and Reviewers,
Ref.: sustainability-1851625
Title: Particle flow analysis on mechanical characteristics of rock with two pre-existing fissures
Thank you very much for giving us this opportunity to revise our manuscript, and we are very grateful to the editor and reviewers for the helpful suggestions. Those comments are all valuable and very helpful for revising and improving our paper, as well as an important guiding significance to our future researches. We have made revisions as commented, as marked in red in the revised submission.
Together with the revised manuscript, a point-by-point response to the editor’s and reviewers’ comments has been attached.
Thanks for your effort and best regards.
Yours sincerely,
Zhenzi Yu, Ang Li*, Bo Zhang, Hongyue Li, Qian Mu, Yonggen Zhou, Shuai Gao
Detail Revisions for ID sustainability-1851625
Title: Particle flow analysis on mechanical characteristics of rock with two pre-existing fissures
Manuscript Number: sustainability-1851625
The authors presented an approach of “Particle flow analysis on mechanical characteristics of rock with two pre-existing fissures.” The subject is interesting and within the scope of the journal: Sustainability. However, I think significant concerns are reasonable and could improve the manuscript’s quality before publication. Overall, I would recommend publication of this manuscript subject to “major revision” −taking into account precisely addressing the following comments in the revised version.
Question 1: 1. The English language does not comply with the standard required for an international journal article.
Response: Thanks for your kind suggestion. We have checked and revised the manuscript according to your suggestion and marked in red in the revised submission. We are sorry for the inconvenience.
Question 2: Abstract has flaws. Please make the abstract precise with the following information: background, methods, results, and concluding remarks.
Response: Thanks for this good suggestion. We have rewritten the Abstract part according to your suggestion and marked in red in the revised submission. Your comments are all valuable and very helpful. And we will pay more attention to this in our future researches.
Question 3: I’d suggest adding model/result validation in the abstract section and the methodology part. Besides, the concluding remarks are not well written. I recommend stressing it in this section.
Response: Thanks for your valuable suggestion. We have added model validation in the methodology part and rewritten the conclusion part.
(a) Stress-strain curves and failure modes (b) Numerical model based on PFC
Fig. 2 Stress-strain curves, failure modes and numerical model of experimental results and numerical results
Question 4: I am not sure what is the main research hypothesis. What are the research gaps?
Response: Thanks for your valuable suggestion. Sorry for this inconvenience. Our research is based on the assumption that rocks are homogeneous materials. But in fact, rocks are not homogeneous materials. The reason why we make this assumption is mainly to study the influence of cracks on the strength, deformation and failure of rocks alone. Next, we will consider the heterogeneity of rock and study the influence of cracks.
Question 5: What are the main contributions of this study?
Response: Thanks for your kind suggestion. There are many researches on fractured rocks. Rock has become a geological body that must be dealt with in rock engineering, such as mining engineering, tunnel engineering, carbon dioxide geological storage, nuclear waste storage and geothermal mining. As a naturally formed geological body, rock mass is composed of various joint fissures and rock blocks cut by fissures. Many research results show that under any stress state, rock mass is most likely to crack, expand, bifurcate and penetrate from the fissure tip, resulting in instability and failure. Therefore, it is of great significance to study the basic characteristics, failure mechanism and mechanical model of fissures for the safety and stability of rock engineering. However, the influence of cracks on rock strength, crack evolution and deformation from the microscopic perspective is relatively small. This paper attempts to explain the effect of cracks on rock mechanical properties from a microscopic point of view by using the discrete element software PFC.
Question 6: Justification of selecting a model is required.
Response: Thanks for your kind suggestion. PFC attempts to explain the mechanical properties and behavior of media from a meso perspective, which has been widely used in rock engineering. In PFC numerical simulation, the choice of model has a great influence on the research results. The bonding between particles will be damaged by external effects, resulting in the separation of particles, so as to simulate the generation and propagation of cracks in the medium. In the process of simulating particle bonding failure, PFC program provides two basic particle bonding models: Contact bonding and parallel bonding. However, the contact bonding model is point contact and can not transmit torque, so the parallel bonding model is more used in rock simulation. And the parallel bonding model has been applied in many papers. [1] Zhao TB, Guo WY, Lu CP, et al. Failure characteristics of combined coal-rock with different interfacial angles[J]. Geomechanics & Engineering, 2016, 11(3):345-359. Yin Dawei, Chen Shaojie, Chen Bing, et al. Strength and failure characteristics of the rock-coal combined body with single joint in coal[J]. Geomechanics and Engineering, 2018, 15(5):1113-1124. In view of the content, length and other factors of the paper, the verification and justification of the model are not repeated in this paper.
Question 7: Discussion is missing. The discussion section needs to be added separately. The contents sometimes jump between topics without a clear direction. I would have wished to see more information on the actual meaning of the findings and how the results add to the broader issue and the specific scientific field.
Response: Thanks for this good suggestion. According to your opinion, we have added some descriptions in the results section and added a discussion section. The conclusion, significance and deficiency of this paper are described in detail in the discussion part. If there are any questions please let us know.
Special thanks to you for your good comments. A point-by-point response to the comments has been made, and the revised manuscript shows the changes highlighted in red. All of your suggestions are very important, and they have important guiding significance for my thesis writing and research work. Your efforts in the review process of this manuscript are greatly appreciated.

Reviewer 3 Report
As considering the subject and results, the study gives some new original information and thus has scientific merit for publication. This paper has enough novelty for publication in this journal after following comments:
1. English of the paper should be polished carefully.
2. Check for few minor typos and punctuation mistakes within the text,
3. The difference of the study (originality of the study) from the studies in the literature and the aim of the study should be given in the introduction section with clear sentences.
4. The introduction can be improved.
https://onlinelibrary.wiley.com/doi/10.1002/zamm.202100287
http://www.techno-press.org/content/?page=article&journal=anr&volume=12&num=4&ordernum=6
Author Response
Sustainability
Responses to comments
Dear Editors and Reviewers,
Ref.: sustainability-1851625
Title: Particle flow analysis on mechanical characteristics of rock with two pre-existing fissures
Thank you very much for giving us this opportunity to revise our manuscript, and we are very grateful to the editor and reviewers for the helpful suggestions. Those comments are all valuable and very helpful for revising and improving our paper, as well as an important guiding significance to our future researches. We have made revisions as commented, as marked in red in the revised submission.
Together with the revised manuscript, a point-by-point response to the editor’s and reviewers’ comments has been attached.
Thanks for your effort and best regards.
Yours sincerely,
Zhenzi Yu, Ang Li*, Bo Zhang, Hongyue Li, Qian Mu, Yonggen Zhou, Shuai Gao
Detail Revisions for ID sustainability-1851625
Title: Particle flow analysis on mechanical characteristics of rock with two pre-existing fissures
Manuscript Number: sustainability-1851625
Comments of the reviewer #3:
As considering the subject and results, the study gives some new original information and thus has scientific merit for publication. This paper has enough novelty for publication in this journal after following comments.
Question 1: English of the paper should be polished carefully.
Response: Thanks for your kind suggestion. We have checked and revised the manuscript according to your suggestion and marked in red in the revised submission. We are sorry for the inconvenience.
Question 2: Check for few minor typos and punctuation mistakes within the text.
Response: Thanks for your good suggestion. We have checked and revised the manuscript according to your suggestion and marked in red in the revised submission.
Question 3: The difference of the study (originality of the study) from the studies in the literature and the aim of the study should be given in the introduction section with clear sentences.
Response: Thanks for your valuable advice. We are sorry for the inconvenience. We rewritten the Introduction according to your opinions and added some relevant references.
Question 4: The introduction can be improved.
Response: Thanks for your valuable advice. We are sorry for the inconvenience. We rewritten the Introduction according to your opinions and added some relevant references.
(1) Erdal Ö, Bahar ŞŞ, Ecren UY, et al. On the plane receding contact between two functionally graded layers using computational, finite element and artificial neural network methods. Journal of Applied Mathematics and Mechanics, 2022, https://doi.org/10.1002/zamm.202100287.
(2) Murat Y. Simulate of edge and an internal crack problem and estimation of stress intensity factor through finite element method. Advances in Nano Research, 2022, 12(4): 405-414.
Special thanks to you for your good comments. A point-by-point response to the comments has been made, and the revised manuscript shows the changes highlighted in red. All of your suggestions are very important, and they have important guiding significance for my thesis writing and research work. Your efforts in the review process of this manuscript are greatly appreciated.

Reviewer 4 Report
Dear Authors,
I have gone through the paper titled "Particle flow analysis on mechanical characteristics of rock 2 with two pre-existing fissures ". I have found paper well written. The paper is about experimental study of fractures/fissures in rock material, which is then backed by numerical investigation. Finite element (discrete element) software is used for simulation of results and carrying out the analysis. Paper is overall well written and informative for people working civil and mining industry. Paper could be extended to 3D for further relevance of the study. Experimental results are well presented and discussion also covers the limitation of the approach.
I have found only few minor errors like authors mention using a software for analysis but not much details are provided. Term PFC is used but never explained what PFC stand for is it name of software or does it related to particle flow model ? This is an important point and should be clarified.
All figures are good expect figure 7, which appears a bit hazy and not of the best quality and should be improved. All figures captions are single liner and does not tell much about what reader should be looking at in the figure and there is room to improve the figures.
If these minor changes are taken care of I would be happy to recommend paper for publication.
Regards
Author Response
I have gone through the paper titled "Particle flow analysis on mechanical characteristics of rock 2 with two pre-existing fissures ". I have found paper well written. The paper is about experimental study of fractures/fissures in rock material, which is then backed by numerical investigation. Finite element (discrete element) software is used for simulation of results and carrying out the analysis. Paper is overall well written and informative for people working civil and mining industry. Paper could be extended to 3D for further relevance of the study. Experimental results are well presented and discussion also covers the limitation of the approach.
Question 1: I have found only few minor errors like authors mention using a software for analysis but not much details are provided. Term PFC is used but never explained what PFC stand for is it name of software or does it related to particle flow model ? This is an important point and should be clarified.
Response: Thanks for your good suggestion. We are sorry for the inconvenience. PFC is particle flow code. We have revised it and marked in red in the revised submission.
Question 2: All figures are good expect figure 7, which appears a bit hazy and not of the best quality and should be improved. All figures captions are single liner and does not tell much about what reader should be looking at in the figure and there is room to improve the figures.
Response: We are sorry for the inconvenience. We have revised it and marked in red in the revised submission.
Question 3: If these minor changes are taken care of I would be happy to recommend paper for publication.
Response: Special thanks to you for your good comments. A point-by-point response to the comments has been made, and the revised manuscript shows the changes highlighted in red. All of your suggestions are very important, and they have important guiding significance for my thesis writing and research work. Your efforts in the review process of this manuscript are greatly appreciated.
Round 2
Reviewer 1 Report
I went through the revised manuscript titled "Particle flow analysis on mechanical characteristics of rock with two pre-existing fissures" written by Zhenzi Yu and others. The manuscript describes about the numerical modeling using the discrete element method (DEM) performed for simulating the rock deformation process with respect to the two pre-existing parallel fissures. Although I found the manuscript improved at some degrees, it is difficult to say that the manuscript reaches enough level to be published in an international journal. The most serious problem of the manuscript is, I should emphasize again, the lack of the review of the previous studies. The authors should compare the present results with the existing knowledge and clearly indicate the importance of the new ones. For example, the authors listed up some literature in the introduction as follows (lines 49-52). In the following sentences, the readers would see there are some existing works dealing with the pre-existing fissures, but they cannot understand what are indicated by previous studies. the previous literature studied the relationship between the number of fissures and deformation process, then how the number of fissures affects the deformation process? Unless such information is presented, the readers cannot understand the implications of the manuscript. "The study attempts to explain the influence of fissures on rocks from light, sound and electric signals [10-13] . The research angles of fissures are also diverse, including whether the fissures are through, the length, width and angle of cracks, as well as the number and location of fissures [14-18]." The discussion section has a similar flaw. The authors emphasized that the rose diagram is provided in the manuscript. This point may be the (only) strength of the present manuscript, although its importance is not shown at all. If the rose diagram of the phenomena is available, what kind of benefit do the readers obtain? They do not show such indication at all. The authors noted as follows: " It can accurately describe the failure direction of microcracks and the number of microcracks in that direction." (line 243), however, the information to employ it effectively, such as accuracy analysis, is not provided.
Author Response
I went through the revised manuscript titled "Particle flow analysis on mechanical characteristics of rock with two pre-existing fissures" written by Zhenzi Yu and others. The manuscript describes about the numerical modeling using the discrete element method (DEM) performed for simulating the rock deformation process with respect to the two pre-existing parallel fissures. Although I found the manuscript improved at some degrees, it is difficult to say that the manuscript reaches enough level to be published in an international journal. The most serious problem of the manuscript is, I should emphasize again, the lack of the review of the previous studies. The authors should compare the present results with the existing knowledge and clearly indicate the importance of the new ones.
Question 1: For example, the authors listed up some literature in the introduction as follows (lines 49-52). In the following sentences, the readers would see there are some existing works dealing with the pre-existing fissures, but they cannot understand what are indicated by previous studies. the previous literature studied the relationship between the number of fissures and deformation process, then how the number of fissures affects the deformation process? Unless such information is presented, the readers cannot understand the implications of the manuscript. "The study attempts to explain the influence of fissures on rocks from light, sound and electric signals [10-13]. The research angles of fissures are also diverse, including whether the fissures are through, the length, width and angle of cracks, as well as the number and location of fissures [14-18]."
Response: Thanks for your kind advice. We are sorry for the inconvenience. We rewritten the Introduction according to your opinions and added some relevant references.
Question 2: The discussion section has a similar flaw. The authors emphasized that the rose diagram is provided in the manuscript. This point may be the (only) strength of the present manuscript, although its importance is not shown at all. If the rose diagram of the phenomena is available, what kind of benefit do the readers obtain? They do not show such indication at all. The authors noted as follows: " It can accurately describe the failure direction of microcracks and the number of microcracks in that direction." (line 243), however, the information to employ it effectively, such as accuracy analysis, is not provided.
Response: The rose diagram of microcrack distribution is obtained in this paper. It can accurately describe the failure direction of microcracks and the number of microcracks in that direction. In the control of engineering fracture and failure of rock with pre-existing fissures, it can be controlled according to the propagation direction of a large number of microcracks. In this way, the waste of manpower and material resources caused by the control technology such as grouting or bolting due to the lack of clear control direction can be avoided. However, it is true that we have not yet provided information on how to effectively use it, and we will study this in detail in our future research.
Special thanks to you for your good comments. A point-by-point response to the comments has been made, and the revised manuscript shows the changes highlighted in red. All of your suggestions are very important, and they have important guiding significance for my thesis writing and research work. Your efforts in the review process of this manuscript are greatly appreciated.
Reviewer 2 Report
The paper is now ready to publish in its current form. Thank for your effort.
Author Response
The paper is now ready to publish in its current form. Thank for your effort.
Response: Thanks. Your efforts in the review process of this manuscript are greatly appreciated.
Reviewer 3 Report
Dear Editor,
This paper is acceptable.
All corrections have been made.
Best regards
Author Response
This paper is acceptable. All corrections have been made.
Response: Thanks. Your efforts in the review process of this manuscript are greatly appreciated.
Round 3
Reviewer 1 Report
My suggestion is, unfortunately, similar to that I sent in the first stage (and second stage) of the review. The problems I pointed out have remained. This is the third time for me to point out that the problem of this manuscript is the lack or deficiency of the review, in other words, background knowledge. Therefore, the readers cannot judge what is new and how it is important. The newly added description (line 49) tells that "the more cracks, the smaller peak strength [9]" referring Dai et al. (2020) that might studied the influence of multiple pre-existing cracks on the rock strength. (By the way, I could not find neither this article nor the journal (working in 2020) on the internet... Is this really existing?) This shall indicate the impact of pre-existing fissures on the rock strength, but simultaneously indicate its complexity, such as number and direction of each fissure have strong effects. Then, the importance of double-parallel-fissure system, which may become a very simple system compared to the natural cases, is missing in the present manuscript. Although including detailed reviews of the relevant area may decrease the importance of the content of this manuscript, such work is mandatory in the article in an international journal, I believe. Above is one example and there are some problems as I did or did not mentioned in the previous stage of the review. As I am not the supervisor of the authors, I would not point out all of the problems. If the editor really believes this manuscript is adequately modified and is worth publishing in the journal, they should decide so without asking those who suggested "reject" and gave up to point out all of the problems, shouldn't they? Below, I attach my first and second review comments even though they are available in the system. [2nd stage] The review request mail tells the due is 3 days; however, I could not believe that an decent publisher proposes such a schedule. Anyway, I went through the revised manuscript titled "Particle flow analysis on mechanical characteristics of rock with two pre-existing fissures" written by Zhenzi Yu and others. The manuscript describes about the numerical modeling using the discrete element method (DEM) performed for simulating the rock deformation process with respect to the two pre-existing parallel fissures. Although I found the manuscript improved at some degrees, it is difficult to say that the manuscript reaches enough level to be published in an international journal. The most serious problem of the manuscript is, I should emphasize again, the lack of the review of the previous studies. The authors should compare the present results with the existing knowledge and clearly indicate the importance of the new ones. For example, the authors listed up some literature in the introduction as follows (lines 49-52). In the following sentences, the readers would see there are some existing works dealing with the pre-existing fissures, but they cannot understand what are indicated by previous studies. the previous literature studied the relationship between the number of fissures and deformation process, then how the number of fissures affects the deformation process? Unless such information is presented, the readers cannot understand the implications of the manuscript. "The study attempts to explain the influence of fissures on rocks from light, sound and electric signals [10-13] . The research angles of fissures are also diverse, including whether the fissures are through, the length, width and angle of cracks, as well as the number and location of fissures [14-18]." The discussion section has a similar flaw. The authors emphasized that the rose diagram is provided in the manuscript. This point may be the (only) strength of the present manuscript, although its importance is not shown at all. If the rose diagram of the phenomena is available, what kind of benefit do the readers obtain? They do not show such indication at all. The authors noted as follows: " It can accurately describe the failure direction of microcracks and the number of microcracks in that direction." (line 243), however, the information to employ it effectively, such as accuracy analysis, is not provided. Consequently, the manuscript is still a kind of the report, and I cannot recommend to publish it in an international journal. [1st stage] Dear the editor, I went through the manuscript titled "Particle flow analysis on mechanical characteristics of rock with two pre-existing fissures" written by Zhenzi Yu and others. The manuscript describes about the numerical modeling using the discrete element method (DEM) performed for simulating the rock deformation process with respect to the two pre-existing parallel fissures. The manuscript lacks appropriate reviews of the existing works, thus the authors failed to explain how their results are important for the common knowledges of rock deformation mechanics. For example, in line 37, "Domestic scholars have done a lot of research on the influence of fissures on rocks, including large-scale in-situ test and laboratory scale fissure research." is stated without any appropriate references. It means that the authors show nothing about the research history of the relevant area. Mainly because of such poor review on the existing works, the this manuscript only succeeded in describing the results of simple 5 numerical experiments without any discussions. The authors described some results, but they did not discuss whether these results matches with existing works or theories. Unfortunately, I should have to say that this manuscript does not reach to the level of students' reports, and thus cannot recommend to publish in an international journal such as Sustainability. Below I note some other point to be modified, though I gave up to pick them all. [specific concerns] Appropriate figure captions are required. The authors provide too simple figure captions such as "Figure 4. Microcrack evolution with axial strain." However, more detailed explanation is required in general. For example, Figure 4b is for the axial angle for 45 degree, which is explained in the text. This kind of explanation is also required in the figure captions, otherwise, readers have difficulty to understand the figures. l60: PFC, which is developed by ITASCA, can be referred following the guideline provided by the company. The authors should appropriately refer the commercial software under their rules. https://www.itascainternational.com/software/faqs/how-should-i-cite-itasca-software-in-my-publication-or-report l81: Figure 1 is not properly shown. l101: maybe the authors had better to show one typical results (such as shown in Figures 7-9) in Figure 2 to explain how to see the results of the present numerical experiments. l163: Figure 7 requires explanation such as "which domain is the rock, fissure and so on" 7 August
Author Response
Comments of the reviewer #1:
My suggestion is, unfortunately, similar to that I sent in the first stage (and second stage) of the review. The problems I pointed out have remained. This is the third time for me to point out that the problem of this manuscript is the lack or deficiency of the review, in other words, background knowledge. Therefore, the readers cannot judge what is new and how it is important.
Response: Thank you very much for giving us another opportunity to revise our manuscript, and we are very grateful to you for the helpful suggestions. We have rewritten the introduction and revised the references accordingly.
Question 1: The newly added description (line 49) tells that "the more cracks, the smaller peak strength [9]" referring Dai et al. (2020) that might studied the influence of multiple pre-existing cracks on the rock strength. (By the way, I could not find neither this article nor the journal (working in 2020) on the internet... Is this really existing?)
Response: We are sorry for the inconvenience. We have rewritten the introduction and revised the references accordingly. What is more, Reference [9]’s DOI: DOI:10.13827/j.cnki.kyyk.2020.01.002. [9] Dai, B. , Wang, J.B. , Zhao, J. , Liu, F. , & Chen, K.X. . (2020). Analysis of the influence of different dip angles and crack numbers on rock strength and crack propagation law. Mining Research and Development, 40(1):7-11.
Question 2: This shall indicate the impact of pre-existing fissures on the rock strength, but simultaneously indicate its complexity, such as number and direction of each fissure have strong effects. Then, the importance of double-parallel-fissure system, which may become a very simple system compared to the natural cases, is missing in the present manuscript. This shall indicate the impact of pre-existing fissures on the rock strength, but simultaneously indicate its complexity, such as number and direction of each fissure have strong effects. Then, the importance of double-parallel-fissure system, which may become a very simple system compared to the natural cases, is missing in the present manuscript.
Response: We are sorry for the inconvenience. We have rewritten the introduction and revised the references accordingly. We are conducting mechanical tests on rock samples at the laboratory scale. And most of the fissure research in the laboratory is to take rock samples with a height diameter ratio of 2:1 according to the standards of the international society of rock mechanics, and then form fissures through hydraulic cutting to study their influence on rock strength, deformation and failure. We added relevant description in the second paragraph of the introduction and relevant references.
Question 3: Although including detailed reviews of the relevant area may decrease the importance of the content of this manuscript, such work is mandatory in the article in an international journal, I believe.
Response: We are deeply sorry for the inconvenience and carelessness. We have rewritten the introduction and revised the references accordingly. The research significance, innovation and research status of related research of the paper are rewritten. More references from international journals have been added.
Question 4: Above is one example and there are some problems as I did or did not mentioned in the previous stage of the review. As I am not the supervisor of the authors, I would not point out all of the problems. If the editor really believes this manuscript is adequately modified and is worth publishing in the journal, they should decide so without asking those who suggested "reject" and gave up to point out all of the problems, shouldn't they?
Response: We are deeply sorry for the inconvenience. We have rewritten the introduction and revised the references accordingly. And we also check and modify the full text. The research significance, innovation and research status of related research of the paper are rewritten. More references from international journals have been added. Thanks.
Special thanks to you for your good comments. A point-by-point response to the comments has been made, and the revised manuscript shows the changes by “Track Changes” function. All of your suggestions are very important, and they have important guiding significance for my thesis writing and research work. Your efforts in the review process of this manuscript are greatly appreciated.